# Sunflower and Palm Kernel Meal Present Bioaccessible Compounds after Digestion with Antioxidant Activity

**DOI:** 10.3390/foods12173283

**Published:** 2023-09-01

**Authors:** Mariana Sisconeto Bisinotto, Daniele Cristina da Silva Napoli, Fernando Moreira Simabuco, Rosângela Maria Neves Bezerra, Adriane Elisabete Costa Antunes, Fabiana Galland, Maria Teresa Bertoldo Pacheco

**Affiliations:** 1CCQA, Science and Quality Food Center, Institute of Food Technology (ITAL), Av. Brasil, 2880, Campinas 13070-178, SP, Brazil; 2LABMAS, Multidisciplinary Laboratory in Food and Health, School of Applied Sciences (FCA), University of Campinas (UNICAMP), Limeira 13484-350, SP, Brazil; 3LLPP, Dairy Products, Probiotics and Prebiotics Laboratory, School of Applied Sciences (FCA), University of Campinas (UNICAMP), Limeira 13484-350, SP, Brazil

**Keywords:** prebiotic potential, amino acids profile, phenolic compounds, antioxidant activity, DNA protection, *Helianthus annuus* L., *Elaeis guineensis* Jacq.

## Abstract

Sunflower (*Helianthus annuus* L.) and African palm kernel (*Elaeis guineensis* Jacq.) are among the most cultivated in the world regarding oil extraction. The oil industry generates a large amount of meal as a by-product, which can be a source of nutrients and bioactive compounds. However, the physiological effects of bioactive compounds in such matrices are only valid if they remain bioavailable and bioactive after simulated gastrointestinal digestion. This study evaluated the chemical composition and antioxidant and prebiotic potential of de-oiled sunflower (DS) and de-oiled palm kernel (DP) meal after in vitro digestion. The DS sample had the highest protein content and the best chemical score, in which lysine was the limiting amino acid. Digested samples showed increased antioxidant activity, measured by in vitro methods. The digested DS sample showed a better antioxidant effect compared to DP. Moreover, both samples managed to preserve DNA supercoiling in the presence of the oxidizing agent. The insoluble fractions after digestion stimulated the growth of prebiotic bacterium, similar to inulin. In conclusion, simulated gastrointestinal digestion promoted in both matrices an increase in protein bioaccessibility and antioxidant capacity, pointing to a metabolic modulation favorable to the organism.

## 1. Introduction

The food waste concept has emerged as an attempt to make better use of edible materials (lost, discarded, or spoiled) collected from the food industry all along the production chain up to consumption in households [1], consistent with the circular economy and the 17 Sustainable Development Goals [2]. Regarding the oilseed world market, sunflower (*Helianthus annuus* L.) and African oil palm (*Elaeis guineensis* Jacq.) are among the most cultivated oilseeds crops, and around 22.3 and 8 million tons of sunflower and palm kernel meal, respectively, will be produced by the edible oil industry in 2023/24 [3]. This massive amount of biomass is enriched in bioactive components [4], which can be further explored. For example, besides its high phenolic compounds content, de-oiled sunflower meal is a favorable source of highly functional and nutritional-quality protein for plant-based products [5]. The de-oiled palm kernel meal has 37% dietary fiber and 18.6% crude protein that also could be extracted for application in other products [6]. Traditionally, de-oiled meals have been applied to animal feed based on their high fiber content. However, other applications rather than feed could be addressed, such as their introduction into the human diet to take advantage of the bioactive compounds [1,7].

Phenolic compounds and bioactive peptides are examples of bioactive components [8,9]. The bioactive component must present bioaccessibility to exercise a biological function (bioactivity). In other words, it must become accessible before being absorbed in the gastrointestinal tract. Therefore, it is necessary to release the component from the food matrix to become soluble [10]. Phenolic compounds are antioxidants based on the presence of aromatic rings in their structure. Furthermore, their intake correlates with a reduction in chronic degenerative disease risk development [8]. A previous study has claimed that in vitro digestion can increase or decrease the total phenolic compound content, depending on the type of food matrix (liquid or solid), since the dietary compounds of the solid matrix can protect their structure. For example, the total phenolic content increased in cereals, legumes, vegetables, chocolates, and fruits and was reduced in wines after in vitro digestion [8]. Food-derived bioactive peptides are sequences of 2 to 20 amino acids, usually inactive in parent protein, that are released by proteolysis of the native protein (for example, in fermentative and proteolytic manufacturing processes, in vivo or in vitro digestion) and can develop bioactive activities, such as lowering blood pressure and cholesterol, anticancer or antioxidant activities, among other functionalities [9]. Sunflower protein hydrolysis has been claimed to release antioxidant peptides [11]. However, the form of the peptide’s resistance to simulated digestive processes has not been evaluated.

Another category of bioactive compounds is prebiotics, whose consumption has also been stimulated to modulate the gastrointestinal microbiota composition toward health benefits [12]. Phenolic compounds may present prebiotic effects after digestion [13]. Some examples of well-established prebiotics are fructans, oligofructose, inulin, fructooligosaccharides (FOSs), lactulose, galactan, galactooligosaccharides (GOSs), resistant starch (RS), pectin, and complex fibers [14]. However, to our knowledge, the prebiotic effect of de-oiled sunflower and palm kernel meal has yet to be investigated.

By elucidating the aspects mentioned above, repurposing sunflower and palm kernel de-oiled meals as a source of biologically active compounds has health and environmental appeal and confers an economic advantage by utilizing materials of diminished commercial value. Nonetheless, their bioactive compounds must be bioaccessible after gastrointestinal digestion. To our current understanding, this is the first research that verifies the antioxidant capability of sunflower and palm kernel de-oiled meal after in vitro digestion. [15]. Hence, the current study assesses the potential antioxidant and prebiotic effects of de-oiled sunflower and palm kernel meal concerning their potential application as bioactive ingredients in human nutrition. Characterizing the digested fraction will facilitate the comprehension of the underlying mechanisms through which these bioactivities may occur.

## 2. Materials and Methods

### 2.1. De-Oiled Flour Preparation

Dehulled and semi-de-oiled meal products of palm kernel and sunflower were supplied by Agropalma, Moju/PA/Brazil and Veris Óleos Vegetais Ltd. a-ME, Vinhedo city/SP state/Brazil, respectively. The residual oil was extracted using Soxhlet equipment (16 h, n-hexane), and then the particle size was reduced to 5 mm. De-oiled palm kernel meal (5 g) was added to 20 mL of sodium acetate buffer (0.2 M, pH 5.4) and autoclaved (1.1 bar, 121 °C, 15 min) so that fibers were loosened, and digestive enzyme action was favored [16].

### 2.2. Chemicals and Reagents

*Bifidobacterium animalis* BLC1 and *Lactobacillus plantarum* BG112 (recently re-classified as *Lactiplantibacillus plantarum* BG112 by Zheng et al. (2020) [17]) were supplied by Sacco (Campinas, SP, Brazil), and *Lactobacillus acidophilus* LA-5 and *Bifidobacterium animalis* BB-12 were supplied by Christian-Hansen (Hoersholm, DNK). Further reagents are described in the Appendix A (Appendix A).

### 2.3. In Vitro Gastrointestinal Digestion (GID)

GID was conducted according to Minekus et al. (2014) [18]. Before starting the oral phase (pH 7, 10 min, alpha amylase), 20 mL of deionized water was added to improve matrix and enzyme contact. Afterward, both gastric (pH 3, pepsin) and intestinal (pH 7, pancreatin) phases were maintained for a 2 h duration. GID was conducted in a water bath under stirring (37 °C, 90 rpm) and stopped by heating (90 °C, 10 min) and cooling down in a cold-water bath (4 °C). The whole digested material was centrifugated (3645× *g*; 30 min; 4 °C) to separate soluble and insoluble fractions, which were freeze-dried and kept at −20 °C for further analyses. A blank of digestion was conducted as described above to be discounted from all the results. 

### 2.4. Chemical Characterization and Phenolic Compounds

Samples were characterized (moisture, protein, lipids, fiber, and ash) according to AOAC methods [19]. The amino acid profile was carried out, as previously described by White, Hart, and Fry (1986) [20], with pre-column derivatization and elution in a high-efficiency liquid chromatograph (Shimadzu Corporation, Kyoto, Japan) with a C18 column LUNA 100 Å (4.6 mm × 250 mm; particle size 5 µm) (Phenomenex, Torrance, CA, USA). The quantification was performed through a standard curve of standard amino acids and DL-2-aminobutyric acid as the internal standard. Tryptophan was determined by the enzymatic method according to Spies (1967) [21]. The amino acid score was calculated according to WHO/FAO/UNU (2007) [22]. To determine total phenolic compounds, protein precipitation followed by reaction with Folin–Ciocalteu reagent was performed, and the results were expressed as mg gallic acid equivalent (GAE)/g sample [23].

### 2.5. Molecular Weight Distribution Profile

The distribution of molecular weight (MW) was performed based on the retention time of the aromatic compounds [24]. Briefly, samples were dissolved (5 mg/mL) in sodium phosphate buffer (25 mM pH 7.4 containing 150 mM NaCl), sonicated for 10 min, and filtered through a 45 µm polytetrafluoroethane membrane before injection (500 µL). Isocratic elution (0.5 mL/min, 90 min) and UV detection at 280 nm were performed on size exclusion fast protein liquid chromatography (SE-FPLC; Akta Pure 25, GE Healthcare, Chicago, IL, USA) with Unicorn 6.3 software and Superdex-200 10/300 GL and Superdex-30 10/300 GL columns (GE Healthcare, Chicago, IL, USA) arranged in series. The α-lactalbumin (14,178 Da), insulin (5807.6 Da), vitamin B12 (1355.37 Da), and L-β-4-dihydroxyphenylanine (197.2 Da) standards were used to construct an analytical curve to define MW ranges.

### 2.6. Extract Preparation for Antioxidant Activities

Samples (100 mg) were mixed with 4 mL of solvent (water or aqueous ethanolic solution (30:70 *v/v*)) and homogenized using an Ultra-Turrax T-25 (IKA, Staufen, Germany) for one minute before centrifugation (1125× *g*, 4 °C, 10 min). The soluble phase was filtered through Whatman no. 2 paper into a 10 mL volumetric flask. This procedure was repeated twice, and the volume was filled up with solvent [25].

### 2.7. Oxygen Radical Absorbance Capacity (ORAC)

ORAC was performed in ethanolic extract samples according to Chisté et al. (2011) [26]. The ability of the samples to scavenge peroxyl radicals (ROO^•^) was tested in terms of their potential to prevent oxidation of the fluorescein molecule. Fluorescence decay was measured during a 2 h reaction at 37 °C in a 96-well microplate fluorescence reader (Synergy, BioTek^®^, Gen5 software) at a wavelength of 485 nm for excitation and 528 nm for emission. AAPH was used as a reactive oxygen species generator (ROS), and a Trolox standard curve (12.5–400 μM) was used to express the results as μmol Trolox equivalent/g sample.

### 2.8. DNA Supercoiled Band Protective Capacity

The prevention of DNA strand breakage from ROS action was assessed in aqueous extract, as described by Yarnpakdee, Benjakul, Kristinsson, and Bakken (2015) [27]. To eliminate any intrinsic proteolytic enzyme, the DS and DP were heated (90 °C, 10 min). A supercoiled plasmid pcDNAFLAG (125 ng/mL, 4 μL) was dissolved in Tris-HCl-EDTA (TE) buffer (10 mM Tris-HCl and containing 0.1 mM EDTA) prepared according to Pavan et al. (2016) [28]. Subsequently, it was mixed to 2 μL of sample aqueous extract containing the samples and 4 μL of aqueous 2.2′-azobis(2-amidino-propane) dihydrochloride (AAPH) 30 mM as oxidizing agent in a DNAse free microtube, using this strict order. The mixture was incubated in darkness at 37 °C for 1 h. Under the same conditions, two DNA controls were performed, one positive and the other negative, replacing the samples (DP and DP) and AAPH by ultrapure water (6 μL and 2 μL), respectively. After incubation, the volume (10 μL) was loaded onto 0.8% agarose gel, and DNA bands were stained with 1:20.000 SYBR safe (Thermo Scientific, Waltham, MA, USA) in Tris-acetate–EDTA (TAE) buffer (40 mM Tris-acetate with 1 mM EDTA). Electrophoresis was conducted at 80 mV for 90 min, followed by 120 mV for 60 min, using a horizontal gel electrophoresis system (Bio-Rad, Hercules, CA, USA). A UV light with the ChemiDoc Imaging System (Bio-Rad, CA, USA) was used to visualize the DNA bands. The quantification was performed using Image J software version 1.53t (NIH, Bethesda, MD, USA). The protective effect of extracts was measured by the retention percentage of supercoiled DNA, calculated according to the following equation.
$$Retention\;supercoiled\;DNA\;band\;\left(\%\right)=\frac{intensit\;of\;sample\;supercoiled\;band}{intensit\;of\;control\;supercoiled\;band}*100$$

### 2.9. ABTS and DPPH Radical Scavenging Assays

The ability to scavenge the 2,2-azinobis(3-ethylbenzothiazoline-6-sulphonic acid (ABTS) and 2,2-diphenyil-picrylhydrazyl (DPPH) radicals was determined according to Al-Duais, Mueller, Boehm, and Jetschke (2009) [29]. The ABTS radical was generated by adding 88 µL of aqueous potassium persulphate (K_2_S_2_O_8_; 140 mM) in 5 mL of aqueous ABTS solution (7 mM). After 16 h in the dark, the absorbance of the ABTS working solution was adjusted to 0.7 ± 0.02 at 734 nm using a UV–Vis spectrophotometer. Then, 20 µL of the sample extract and 220 µL of the ABTS working solution were added to a 96-well microplate. The blank consisted of 240 µL of phosphate buffer (PSB; 75 mM; pH 7.4). A Trolox standard curve was prepared in PBS, and samples were read after six minutes. For the test of the 2,2-diphenyil-picrylhydrazyl (DPPH) radical scavenging capacity, 134 µL of ethanolic DPPH solution (150 µM) and 66 µL of the sample extract were used. A control was performed by replacing the sample extract with 66 µL of ethanol. The blank experiment was performed with 200 µL of ethanol. A Trolox standard curve in ethanol was prepared. The reaction mixtures were kept in the dark for 45 min before reading the absorbance. Absorbance in the ABTS and DPPH assays was read at 730 nm and 517 nm, respectively, using a UV–Vis microplate Synergy Reader (BioTek^®^, Gen5 software). Results were expressed as µmol Trolox equivalent/g sample.

### 2.10. Potential Prebiotic Effect

The ability to stimulate the growth of potentially probiotic bacteria (*Lactobacillus plantarum* BG112, *Lactobacillus acidophilus* LA −5, *Bifidobacterium animalis* BLC1, and *Bifidobacterium lactis* BB −12) was tested in the medium of De Man, Rogosa, and Sharpe (MRS) formulated by its individual components (Appendix A) in such a way that the carbohydrate content (dextrose) was controlled [30]. Sodium acetate (1 g), agar (3 g), dibasic ammonium citrate (0.4 g), peptic digest of animal tissue (peptone A, 2 g), beef extract (2 g), yeast extract (1 g), potassium phosphate (0.4 g), magnesium sulfate (0.02 g), manganese sulfate (0.01 g), 200 μL polysorbate 80, and dextrose (4 g) were diluted in 200 mL deionized water. Inulin (1.89 g) was added to the prebiotic control instead of dextrose, and the same amount of inulin, based on chemical composition, was added to DSi (4 g) and DPi (2.14 g). All MRS media were autoclaved. The freeze-dried bacteria were resuspended in peptone water 0.1% (*w/v*) serial dilution. *Lactobacillus/Lactiplantibacillus* incubation (72 h, 37 °C, microaerophilia) and *Bifidobacterium* incubation (72 h, 37 °C, anaerobiosis by Anaerogen^®^ addition) were performed, and bacterial growth was determined using the spread plate technique, expressed as the log of colony forming units (cfu)/g.

### 2.11. Statistical Analysis

All results were expressed as means ± standard error of the mean (SEM). One-way ANOVA followed by the Dunnett or Kruskal–Wallis test, and the *t*-test for dependent variables or the Wilcoxon test, according to the normality assumption evaluated by the Shapiro–Wilk test, was performed using GraphPad Prism version 9.5.1 for Windows (GraphPad Software, San Diego, CA, USA).

## 3. Results

### 3.1. Chemical Characterization and Total Phenolic Compounds

Chemical characterization (moisture, protein, lipids, and ash) was determined according to the methodologies recommended by the Association of Official Analytical Chemist Methods [19] and is presented in Table 1. It is vital to consider that the composition reflects the centesimal sum of the components, which justifies the increase of some components with the decrease of others. Regarding the macro components, de-oiled sunflower meal (DS) is a better source of protein (52.4 g/100 g) than de-oiled palm kernel meal (DP) (7.7 g/100 g), which is enriched in fibers (85.5 g/100 g). After in vitro gastrointestinal digestion (GID), the protein content of DP became soluble, but the fiber remained with the insoluble fraction, suggesting that it was poorly digested by gastrointestinal enzymes, being suitable for fermentation by probiotic bacteria. The third and more soluble component of the de-oiled sunflower meal digested soluble fraction (DSs) and autoclaved de-oiled palm kernel meal digested soluble fraction (DPs) was the phenolic compounds, showing that GID was able to increase its bioavailability around 1.3- and 3.6-fold, respectively.

GID did not compromise the amino acid content of the samples (Table 2). Moreover, the protein score showed that both samples have the essential amino acids, except for lysine at DS and lysine, methionine, and cysteine at DP. Thus, DS and DP protein are digested by gastrointestinal enzymes, being a cheap source of protein. It is well-established that antioxidant capacity is related to amino acid hydrophobicity [31]. Therefore, it is also important to highlight that DSs and DPs have 31.55% and 29.55% of hydrophobic amino acids, respectively, based on methionine, phenylalanine, tyrosine, tryptophan, cysteine, and histidine content. Tryptophan was destroyed by acid hydrolysis performed with the amino acid method [20]. Therefore, it was determined by enzymatic reaction with Pronase E, and, subsequently, a colorimetric reaction was performed for quantification [21]. Thus, this amino acid was not quantified in the initial samples (DS and DP), probably due to the fibrous complexity of the matrices, which prevented the enzyme (Pronase E) from accessing tryptophan. In the digested samples (DSs and DPS), tryptophan was more accessible to the enzyme and thus could be quantified spectrophotometrically.

### 3.2. Molecular Weight (MW) Distribution by FPLC-SE

GID improved the DS and DP compounds’ solubilities, as shown by the rise in the chromatograms’ total areas (Table 3). In addition, reductions on the components’ sizes were observed. For instance, at the range of compounds with MW higher than 7 kDa, a 33.14% reduction was observed between DS and DSs, as was an increase of 25.5% of MW between 0.1 and 1 kDa. However, the opposite effect was noticed in DP and DPs, increasing the MW at 7 kDa after digestion. The size reduction suggests that under the analysis conditions, DP compounds were poorly soluble. Gastrointestinal enzymes could digest DP, but a significant fraction of DPs was primarily compounds bigger than 7 kDa.

### 3.3. Antioxidant Activity by ORAC, ABTS, DPPH, and DNA Supercoiled Band Protective Capacity

As shown in Figure 1, GID improved the antioxidant activity of DS and DP with all methods. These results suggest that DS and DP might be antioxidant sources to be included in food products as bioactive ingredients. According to the ORAC assay results, the DSs and DPs antioxidant capacities were 38.3% and 635.2% higher than those of DS and DP, respectively, and the DSs presented an antioxidant power 238-fold higher than that of DPs (1009.6 ± 4.4 and 297.9 ± 2.0 µmol Trolox equivalent/g sample). Digested samples presented higher antioxidant capacities by the ABTS assay than by the DPPH assay. However, no difference was observed between DP and DPs samples with the DPPH method. Regardless of the mechanism of the in vitro assay, all the results have increased the evidence that GID raises the bioactive compounds’ accessibility and antioxidant capacity. According to DNA protective capacity analysis (Figure 2), DS and DSs presented similar bioactivity, maintaining 71.55% and 74.02% of the original DNA band, respectively. The DPs showed 92.65% of the original DNA band protection. Therefore, both antioxidant capacity values in SDs and PDs samples were higher than those of the original meal, showing the positive impact of GID in terms of increasing the bioaccessibility of compounds that had been entrapped in food matrixes. 

### 3.4. Potential Prebiotic Effect

The de-oiled sunflower meal (SDi) and the de-oiled autoclaved palm kernel meal (PDi) insoluble fraction were tested regarding their potential prebiotic effect as a source of carbon to stimulate probiotic/potentially probiotic bacteria growth (Figure 3) versus the control (inulin). 

## 4. Discussion

The physical and chemical composition of de-oiled sunflower and palm kernel meal have shown that DS presents a better protein content (52.41 ± 0.02) compared to DP, representing a good source ingredient for protein-derived products. Nevertheless, DP is enriched in carbohydrates (85.54%) and has been applied to feed in association with soybean and corn [32]. After digestion, the protein bioaccessibility was increased in both samples since DSs and Ds showed higher protein content than the original de-oiled meal. Despite the lower protein content, the solubility more than doubled in the DPs sample. An increase in protein content in the soluble fraction was also observed in a previous study with plant protein isolates (garden peas, grass peas, soybean, and lentils) after digestion protocol [33]. Plant proteins are known to present less digestibility than animal proteins because of the presence of cell walls and anti-nutritional factors. Despite this, the fractionation seems to favor better accessibility of plant-derived protein, making it more accessible to the action of digestive enzymes.

Regarding amino acid composition, sunflower protein is a high-quality protein source since it meets most criteria for indispensable amino acids, except for lysine, indicated by WHO/FAO/UNU. At the same time, palm kernel is limited to lysine, methionine, and cysteine. These results are consistent with preview data for sunflower [34] and for palm kernel [35] isolates. An alternative that has been given to increase the nutritional value of plant protein with a lack of sulfur-containing amino acids, such as methionine and cysteine, is the mixture of two plant protein sources that present complementary amino acids to compensate for the lack of limiting amino acids [36]. The amino acid content is intrinsically related to bioactivity [31].

The bioactivity of food peptide hydrolysates has been extensively explored [11,37]. Different processing methods for protein hydrolysis have been applied in sunflower and palm kernel cake by-products, demonstrating critical biological activities in human health, such as antioxidant activity [38,39]. However, few studies have evaluated the resistance in the biological properties of these peptides after digestion [40]. In this work, it was observed that from integral proteins, the formation of peptides during digestion was sufficient to improve the antioxidant potential of the sample. Therefore, the use of de-oiled sunflower cake as a supplement in foods, without the need for prior hydrolysis, can generate antioxidant properties with a crucial physiological effect on the digestive tract. The same may have happened to phenolic compounds. 

The content of phenolic compounds expressed in gallic acid equivalent per gram of sample is higher in supernatants of gastrointestinal digests obtained by in vitro simulated digestion. Two hypotheses can be listed to justify this result. At first, it can be inferred that the simulated digestion of defatted flour allowed for greater extraction of phenolic compounds present in the matrix, solubilizing them and releasing them to be potentially absorbed in the intestine, and thus migrating to the supernatant of the gastrointestinal digest. In a second hypothesis, it can be inferred that digestion alters the structure of phenolic compounds, allowing new configurations with different sites to emerge. Such sites can be oxidized by the Folin–Ciocalteu reagent in another way, enhancing a possible antioxidant action in the body if absorbed. On top of that, phenolic compounds have been described as radical quenching by electron transfer or proton donation [41] and may be effective against peroxyl radicals released from the thermal degradation of AAPH. Previous studies by our group have identified and quantified chlorogenic, caffeic, coumaric, and vanillic acid in de-oiled sunflower extracts. The chlorogenic acids represented 66.2% of the total phenolics in the extracts, being confirmed as the main phenolic compounds of sunflower meal [42].

As a sensible in vitro screening for antioxidant potential prospection, we observed using the ORAC assay that DS presented better antioxidant capacity than DP. A large part of the antioxidant activity observed in DS may be related to the higher content of phenolic compounds present in the sample. In fact, sunflower meal has up to 4% of its mass in phenolic compounds, and the major component is chlorogenic acid [38,43,44]. Although a low antioxidant capacity of DP was observed, several compounds have been identified in palm kernel meal, such as pyrogallol (1550 μg/g), 4-hydroxybenzoic acid (980 μg/g), gallic acid (590 μg/g), and ferulic acid (560 μg/g), as well as catechol, homovanillyl alcohol, and catechin [43]. The ORAC assay uses peroxyl radical, which is better for antioxidant reactions, and its main antioxidant mechanism is electron transfer (ET) [45]. On the other hand, although ABTS and DPPH methods have been extensively used, they are based on the reactions of radicals that do not happen in biological organisms, being also susceptible to several analysis interferences [46]. The ABTS method identifies antioxidant compounds by electron transfer (ET) and hydrogen atom transfer (HAT) mechanisms, but the DPPH method identifies bioactivity mainly by HAT [45]. Therefore, a highly active antioxidant compound with the ABTS method usually shows lower antioxidant capacity than with the DPPH method [47].

Regarding the DNA protective assay, DSs and DPs partially protect the supercoiled DNA band. Leonard et al. (2006) [47] suggested that ROS damage on DNA molecules could be affected by the scavenging of radicals formed during reaction or by inhibiting radical generation. These might have been the antioxidant mechanisms. Dietary antioxidants may help to maintain the body’s redox homeostasis, avoiding oxidative stress and cellular macromolecule damage, which is connected to mutagenesis and chronic disease development [48]. Furthermore, 57.68 ± 4.59% of the MWs of the total compounds of PDs were greater than 7 kDa. These large compounds may have promoted the steric protection of the DNA molecule against peroxyl radicals, as observed with metallothioneins (from 6 to 7 kDa) and some dietary fibers that, depending on their physicochemical properties, can present binding activity that hinders DNA and epithelial cell damage [1,49]. Likewise, sunflower and palm kernel de-oiled meals could be used as antioxidant ingredients.

Despite the DSi and DPi fiber content, probiotic bacteria growth was similar to inulin control. The major palm kernel carbohydrates are mannoses (78%), cellulose (12%), and glucuronoxylans and arabinoxylans (6%) linked by the β1→4 glycosidic band, which make them resistant to human gastrointestinal enzymes and potentially fermentable by probiotic bacteria. Our results did not corroborate a previous study made by Bello and collaborators (2018) [50]. They observed, in palm kernel samples, a prebiotic effect on *Lb. plantarum ATCC* 8014 and *Lb. rhamnosus* ATCC 53103 growth in MRS medium. The main difference was the addition of isolated polysaccharides rather than whole meal. The availability of protein in MRS also could help in bacteria growth, even if it is a non-prebiotic compound [13]. Categorizing fibers as prebiotics is a difficult goal because it depends on the target host and site. For example, cellulose is a prebiotic in ruminants but not in humans, and xylitol is a prebiotic in the oral cavity but has not been shown to be prebiotic elsewhere [13]. Therefore, SDi and SPi should be further studied as potential prebiotic sources. Dietary management aiming to control health disorders and gut microbiota modulation becomes a usual approach and moves the food industry forward in functional product development [51].

## 5. Conclusions 

Sunflower and palm kernel de-oiled meal are interesting sources of antioxidant compounds released from the food matrix by in vitro gastrointestinal digestion, such as peptides and phenolic compounds. These compounds were effectively solubilized by enzymatic action and had improved antioxidant capacity via ORAC, ABTS, and DNA supercoiled band protection. Moreover, polysaccharides stimulated potential probiotic bacteria as much as the inulin control. Further studies should be carried out to investigate bioaccessibility in the Caco-2 cell model to confirm these compounds’ absorption and bioaccessibility properties, which will contribute to valorizing and upcycling edible oil co-products. 

## Figures and Tables

**Figure 1 foods-12-03283-f001:**
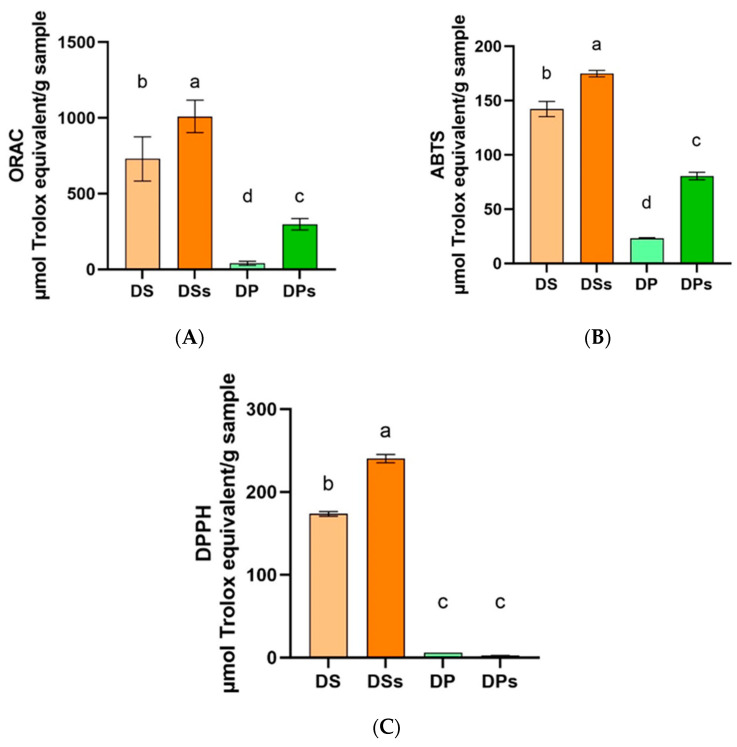
Antioxidant capacity. (**A**) Antioxidant capacity by the ORAC method, (**B**) antioxidant capacity by the ABTS method, and (**C**) antioxidant capacity by DPPH assays. Abbreviations: de-oiled palm kernel meal (DP), de-oiled sunflower meal (DS), digested soluble fraction (PDs), de-oiled sunflower meal digested soluble fraction (SDs). Values were analyzed by one-way ANOVA and Tuckey test (*p* < 0.05). ^a,b,c,d^ superscript lowercase letters means statistical difference (*p* < 0.05).

**Figure 2 foods-12-03283-f002:**
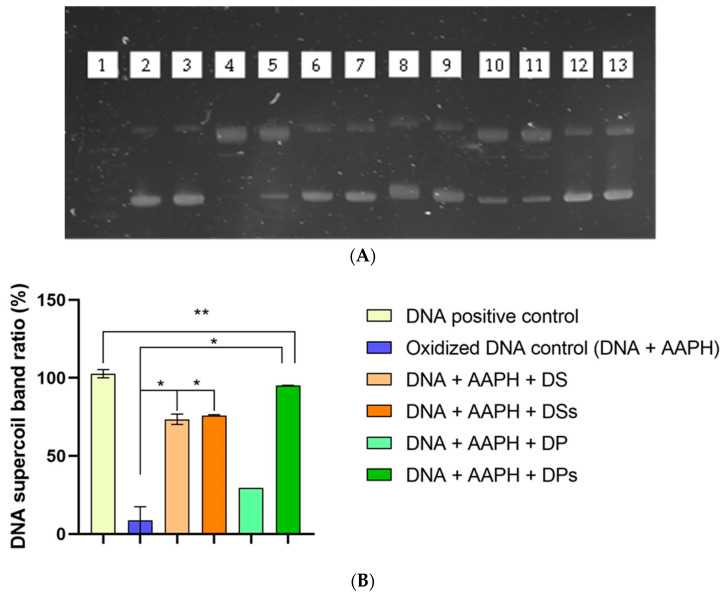
DNA protection and ORAC assays of sunflower and palm kernel de-oiled meal samples. Abbreviations: de-oiled palm kernel meal (DP), de-oiled sunflower meal (DS), digested soluble fraction (PDs), de-oiled sunflower meal digested soluble fraction (SDs). (**A**) Agarose gel electrophoresis, the numbers of which refer to following: 1: molecular weight standard; 2 and 3: DNA positive control; 4 and 5: oxidized DNA control (DNA + AAPH); 6 and 7: DNA + AAPH + DS; 8 and 9: DNA + AAPH + DSs; 10 and 11: DNA + AAPH + DP; 12 and 13: DNA + AAPH + DPs. (**B**) Relationship between DNA positive control and other treatments. (*) Different at *p* < 0.05. (**) No difference at *p* < 0.05.

**Figure 3 foods-12-03283-f003:**
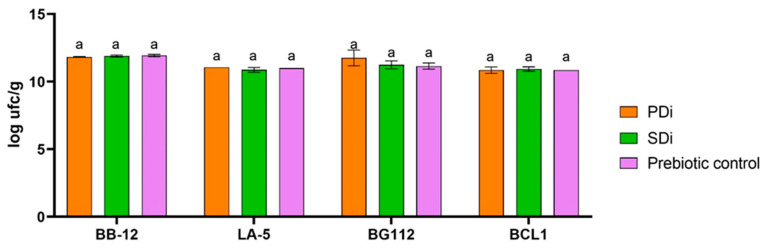
Potential prebiotic effect. Abbreviations: de-oiled palm kernel meal digested insoluble fraction (PDi), de-oiled sunflower meal digested insoluble fraction (SDi) added in the MRS medium as carbon source. Prebiotic control is with inulin addition in the MRS medium. Values were analyzed by one-way ANOVA and Dunnett’s test (*p* < 0.05). ^a^ superscript lowercase letters means there is no statistical difference.

**Table 1 foods-12-03283-t001:** Chemical characterization of sunflower and palm kernel samples, in dry matter.

	DS	DSs	DSi	DP	DPs	DPi
Lipids ^1^	0.4 ± 0.0	nd	nd	0.4 ± 0.0	nd	nd
Protein ^1^	52.4 ± 0.0 ^A^	63.7 ± 0.1 ^A^	36.8 ± 0.2 ^A^	7.7 ± 0.2 ^b^	23.7 ± 0.0 ^a^	7.4 ± 0.0 ^b^
Ash ^1^	8.8 ± 0.1 ^A^	7.2 ± 0.0 ^B^	6.5 ± 0.0 ^C^	2.8 ± 0.0	23.6 ± 0.1 *	3.0 ± 0.0
Carbohydrates ^1^ (*)	38.5	29.1	56.7	89.1	52.7	89.7
Fiber ^1^	18.00 ± 0.4 ^B^	nd	32.6 ± 0.2 ^A^	85.5 ± 0.1 ^a^	nd	85.5 ± 0.1 ^a^
Total phenolic compounds ^2^	17.7 ± 0.9 ^B^	22.9 ± 0.9 ^A^	nd	2.3 ± 0.1 ^b^	8.3 ± 1.0 ^a^	nd

Abbreviations: de-oiled sunflower meal (DS), de-oiled sunflower meal digested soluble (DSs) and insoluble (DSi) fractions, de-oiled palm kernel meal (DP), de-oiled palm kernel digested soluble (DPs) and insoluble (DPi) fractions, not determined (nd). ^a,b^ = lower case letters are related palm kernel; ^A,B,C^ = capital letters are related palm kernel; ^1^ g/100 g; ^2^ eq. Trolox µmol/g sample; (*) carbohydrate was determined by difference.

**Table 2 foods-12-03283-t002:** Amino acids profiles of sunflower and palm kernel de-oiled meal samples and their chemical acid scores.

AA (mg/g Protein)		mg/g ref. Protein *	DS		DSs		DP		DPs	
				Score		Score		Score		Score
**Indispensable ***	Lys	45	38.83 ± 0.04 ^A^	0.86	36.91 ± 0.02 ^A^	0.82	28.56 ± 0.01 ^b^	0.63	36.07 ± 0.21 ^a^	0.80
	Trp	6	nd	0.00	6.37 ± 0.00	1.06	nd	0.00	6.74 ± 0.18	1.12
	Phe + Tyr	38	74.36 ± 0.08 ^A^	1.96	74.22 ± 0.09 ^A^	1.95	80.51 ± 0.08 ^a^	2.12	64.83 ± 0.33 ^b^	1.71
	Met + Cys	22	38.19 ± 0.02 ^A^	1.74	29.61 ± 0.07 ^B^	1.35	19.87 ± 0.01 ^b^	0.90	31.80 ± 0.33 ^a^	1.41
	Thr	23	35.93 ± 0.04 ^B^	1.56	37.71 ± 0.03 ^A^	1.64	34.90 ± 0.02 ^b^	1.52	37.85 ± 0.37 ^a^	1.65
	Leu	59	64.24 ± 0.55 ^A^	1.09	59.32 ± 0.03 ^B^	1.01	74.76 ± 0.01 ^a^	1.27	56.32 ± 0.49 ^b^	0.95
	Ile	30	45.56 ± 0.48 ^A^	1.52	41.43 ± 0.06 ^B^	1.38	38.92 ± 0.07 ^a^	1.30	38.41 ± 0.33 ^a^	1.28
	Val	39	58.22 ± 0.05 ^A^	1.49	50.16 ± 0.22 ^B^	1.29	56.86 ± 0.16 ^b^	1.46	51.09 ± 0.33 ^a^	1.31
	His	15	27.12 ± 0.14 ^A^	1.81	26.16 ± 0.03 ^B^	1.74	15.91 ± 0.01 ^b^	1.06	17.62 ± 0.18 ^a^	1.17
**Dispensable ****	Asp		106.66 ± 0.14 ^B^		111.15 ± 0.07 ^A^		87.11 ± 0.52 ^b^		112.79 ± 0.78 ^a^	
	Glu		220.45 ± 0.28 ^B^		228.77 ± 0.18 ^A^		194.86 ± 0.03 ^b^		198.01 ± 0.02 ^a^	
	Ser		46.28 ± 0.15 ^B^		48.15 ± 0.05 ^A^		54.50 ± 0.01 ^b^		55.35 ± 0.49 ^a^	
	Arg		90.81 ± 0.23 ^B^		92.72 ± 0.03 ^A^		149.74 ± 0.27 ^a^		121.40 ± 1.09 ^b^	
	Ala		44.48 ± 0.12 ^A^		43.17 ± 0.13 ^A^		52.71 ± 0.03 ^a^		45.11 ± 0.40 ^b^	
	Pro		38.32 ± 0.12 ^B^		45.27 ± 0.12 ^A^		46.16 ± 0.08 ^b^		47.79 ± 0.52 ^a^	
	Gly		57.94 ± 0.28 ^B^		68.88 ± 0.16 ^A^		64.63 ± 0.04 ^b^		79.51 ± 0.71 ^a^	
**AA** **distribution**	Hydrophobic		32.70%		31.55%		33.49%		29.55%	
	Hydrophilic		48.39%		49.57%		47.62%		48.59%	
	Neutral		18.91%		18.88%		18.89%		21.86%	

Protein was determined by Kjeldhal [19] (*N* = 5.25 for sunflower and 6.25 for palm kernel). Abbreviations: AA: amino acid; * according to WHO/FAO/UNU [22]; ** without recommendation; nd: not detected; Asp: aspartic acid; Ala: alanine; Arg: arginine; Gln: glutamine; Gly: glycine; His: histidine; Cys: cysteine; Ile: isoleucine; Leu: Leucine; Lys: Lysine; Met: Methionine; Phe: Phenylalanine; Pro: Proline; Ser: Serine; Thr: threonine; Trp: tryptophan; Glu: glutamic acid; Tyr: tyrosine; Val: valine. Hydrophobic (Ala, Val, Met, Phe, Leu, Ile, Pro, Trp), hydrophilic (Arg, Asp, His, Lys, Glu), and neutral AA (Ser, Gly, Thr, Tyr, Cys). Score to adults > 18 years [22]. A *t*-test was performed on each amino acid and sample treatment. Different superscript capital letters and small letters at the same raw are different at *p* < 0.05 in de-oiled sunflower and palm kernel samples, respectively.

**Table 3 foods-12-03283-t003:** Molecular weight distribution of sunflower and palm kernel de-oiled meal samples by FPLC-SE.

Molecular Weight Distribution (Area %)	DS	DSs	DP	DPs
>7 kDa	48.3 ± 4.3 ^A^	16.02 ± 2.4 ^B^	10.28 ± 0.4 ^b^	57.68 ± 4.5 ^a^
5–7 kDa	1.5 ± 0.7 ^A^	3.88 ± 1.8 ^A^	2.88 ± 0.2 ^a^	2.73 ± 1.0 ^a^
3–5 kDa	2.7 ± 1.2 ^A^	6.99 ± 2.2 ^A^	5.36 ± 0.3 ^a^	4.52 ± 1.2 ^a^
1–3 kDa	6.3 ± 3.6 ^A^	6.42 ± 1.9 ^A^	26.64 ± 0.7 ^a^	4.12 ± 1.2 ^b^
0.1–1 kDa	41.2 ± 3.0 ^B^	66.7 ± 4.9 ^A^	54.83 ± 1.0 ^a^	30.95 ± 3.3 ^b^
Total Area (mAU*min)	392.1 ± 6.5 ^B^	1094.5 ± 6.5 ^A^	99.93 ± 1.3 ^b^	2010.48 ± 6.0 ^a^

Abbreviations: de-oiled sunflower meal (DS), de-oiled sunflower meal digested soluble (DSs) fraction, de-oiled palm kernel meal (DP) and de-oiled palm kernel digested soluble (DPs) fractions. A *t*-test for was performed on each range of MW and sample treatment. Different superscript capital letters and small letters at the same raw are different at *p* < 0.05 in de-oiled sunflower and palm kernel samples, respectively.

## Data Availability

The data used to support the findings of this study can be made available by the corresponding author upon request.

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
