# Peer review of "Sunflower and Palm Kernel Meal Present Bioaccessible Compounds after Digestion with Antioxidant Activity"

_foods, 2023, doi:10.3390/foods12173283_

Round 1

Reviewer 1 Report

I have mixed feelings about the article, on the one hand, the research is interesting and raises an extremely important aspect, which is the use of post-production waste in nutrition, on the other hand, it is extremely difficult to read and I have the impression that there is a lot of chaos in it. The lack of line spacing makes it difficult to assess. Have the requirements of the journal changed, or have the authors forgotten to make it available?

Why is the subchapter "preparation of the extract" was numbered 2.6? Was the extract prepared in the same way for all analyses, or only for antioxidant activity and the subsequent points below? If the preparation of extracts was done in the same manner, this part should have been earlier in the article.

Part of the results has not been subjected to statistical analysis. This should be corrected.

Table caption 2: ,,Protein determinated by Kjedhal.” – there is no literature reference. This sentence also contain mistakes: the word determinated should be corrected to determined, and no Kjedhal, but Kjeldahl. As I understand it, the total protein content was determined in the samples, and then the content of individual amino acids was calculated with respect to the "total protein content," but not the weight of the sample?

In many places in the article, numbers are written with a comma instead of a period. This should be corrected.

It's  unfortunate  that the research was based only on the total content of phenolic compounds, and the changes in the content of individual phenolic compounds were not analyzed. I recommend conducting such studies in the future. The disadvantage of the method with the Folin  Ciocalteu reagent is the possible interference with other compounds present in the analyzed sample, such as proteins.

Some citations contain unnecessary numbers. This should be improved.

Author Response

Q) I have mixed feelings about the article, on the one hand, the research is interesting and an extremely important aspect, which is the use of post-production waste in nutrition, on the other hand, it is extremely difficult to read and I have the impression that there is a lot of chaos in it. The lack of line spacing makes it difficult to assess. Have the requirements of the journal changed, or have the authors forgotten to make it available?

R) We are grateful for your comments and have reviewed the manuscript.

We apologize for the lack of line spacing, but it was adopted from the manuscript files format. “Authors are encouraged to use the Microsoft Word template or LaTeX template to prepare their manuscript.”

Q) Why is the subchapter "preparation of the extract" was numbered 2.6? Was the extract prepared in the same way for all analyses, or only for antioxidant activity and the subsequent points below? If the preparation of extracts was done in the same manner, this part should have been earlier in the article.

R) Considering this comment, the title mentioned above was modified. The "preparation of the extract" topic, which is numbered 2.6, only considers the antioxidants activities. Therefore, the name has been completed, as follows: “Preparation of the extract for antioxidants activities”. 

Part of the results has not been subjected to statistical analysis. This should be corrected.

The statistical analysis has been carried out regarding Tables 1 and 2.

Q) Table caption 2: “Protein determinated by Kjedhal.” – there is no literature reference. This sentence also contain mistakes: the word determinated should be corrected to determined, and no Kjedhal, but Kjeldahl. As I understand it, the total protein content was determined in the samples, and then the content of individual amino acids was calculated with respect to the "total protein content," but not the?

The authors are grateful for this observation. The word “determinated” and the name Kjeldhal have been corrected and the literature reference added for the latest. 

We would like to explain that the individual amino acid quantity considered protein content in the sample, using the same unit as the FAO standard for comparison.

In many places in the article, numbers are written with a comma instead of a period. This should be corrected.

The authors are grateful for this observation and we have replaced the commas by dots.

Q) It’s unfortunate  that the research was based only on the total content of phenolic compounds, and the changes in the content of individual phenolic compounds were not analyzed. I recommend conducting such studies in the future. The disadvantage of the method with the Folin  Ciocalteu reagent is the possible interference with other compounds present in the analyzed sample, such as proteins.

R) In previous works by our group, we analyzed the content of individual phenolic compounds and it was verified that the predominant phenolic was chlorogenic acid, which represents about 66 % of the total phenolics. This content and reference were inserted in the manuscript on lines 407- 410.

“Previous studies by our group have identified and quantified chlorogenic, caffeic, coumaric, and vanillic acid in de-oiled sunflower extracts. The chlorogenic acids represented 66.2% of the total phenolics in the extracts, being confirmed as the main phenolic compound of sunflower meal [42].”

Regarding the dosage of total phenolics by the Folin method, it should be clarified that the protein was considered an interference factor and for this reason a control sample was conducted along with the experiment, in order to reduce the interference. The polyphenol extracts (300 μL) were added to 60 μL of 2,2,2-trichloroacetic acid (TCA) aqueous solution (10% w/v) as described by Bisinotto et al., 2021 (this reference was included).

References:

Bisinotto, M.S.; Napoli, D.C.S.; Fino, L.C.; Simabuco, F. M., Bezerra, R. M. N.; Antunes, A. E.C., Pacheco, M. T. B. Bioaccessibility of cashew nut kernel flour compounds released after simulated in vitro human gastrointestinal digestion. Food Research International 2021, 139, 1-8. https://doi.org/10.1016/j.foodres.2020.109906

FRIOLLI, M.P.S.; SILVA, E.K.; NAPOLI, D.C.S.; SANCHES, V.L.; ROSTAGNO, M.A.; PACHECO, M.T.B. High-intensity ultrasound-based process strategies for obtaining edible sunflower (Helianthus annuus L.) flour with low-phenolic and high-protein content. ULTRASONICS SONOCHEMISTRY, 97, 2023, 106449, doi.org/10.1016/j.ultsonch.2023.106449

Q) Some citations contain unnecessary numbers. This should be improved.

R) The reference list has been checked and improved

Reviewer 2 Report

The manuscript entitled “ Sunflower and palm kernel meal present bioaccessible compounds after digestion with antioxidant activity” refers to by-products generated by oil industry which can be used as a source of nutrients and bioactive compounds. In the manuscript, the authors evaluated the chemical composition of de-oiled sun-flower and palm kernel meal, their antioxidant, and prebiotic potential  as well as bioaccessibility of selected components  after in vitro digestion. The conducted research are very important in the point of view the sustainable development and improving human food security by reuse of oil industry by-product. The obtained results are interesting taking into account the practical significance of the considered issue. The provided results fully illustrate the problem under consideration.
The Authors made also statistical analysis of the obtained results, it makes them more valuable. All tests documented in the experimental part are described in a complete, legible manner, enabling verification of the procedures used and their possible replication. The presented results are discussed in the context of a literature review and contain a clear interpretation of the observed changes.

Author Response

The authors are grateful for the observations made by the reviewer and for his kindness in carefully reading the manuscript.

Reviewer 3 Report

The manuscript entitled “sunflower and palm kernel meal present bioaccessible compounds after difestion with antioxidant activity” presents their study of valorization of low value agrobyproducts. The results are interesting and will be of many interests from the valorization field. Meanwhile their presentation is not very effective, and the reviewer needed to stop reading and revisiting what are their arguments often. The reviewer recommends the manuscript after extensive revision of the manuscript.

1.     Title: From the title, the unclearness of story is obvious. After reading the manuscript, the reviewer found what they like to say was “The effects of simulated gasto-intestinal digestion on bioactive compounds and prebiotic properties of sunflower and palm kernel meal”. At the beginning, the reviewer thought “how can they digest meals with antioxidant activity?. Similar story unclearness are found here and there. The authors need to revise the manuscript to make the story clear to EVERYONE.

2.     Use of abbreviations in the Abstract: The abstract contains the abbreviations such as ORAC, ABTS, DDPH, AAPH, ROS, and so on. Many of them are used without definition. Abstracts should be understandable without other part of manuscript as it can be separately representing the article when they are databased.

3.     The story of the Abstract: The story of abstract does not have clear flow. Each item is presented without clarifying the relations between. Meanwhile unnecessary information, such as methods of antioxidant assay, and probiotics tested are included. The author should revise the manuscript with the idea what is important and what are supportive information.

4.     The story of Introduction: From the third paragraph in the introduction, the authors explain the background of what types of valorization will be handled in this manuscript. After reading the manuscript, items presented here are somewhat unclear for the story. I would rather bring the story to focus on GID effects. They give the emphasis on prebiotics activities, but it was not significant. If the story is focused on how GID can affect, then the story becomes clearer. Moreover, the DNA protective capacity is not discussed here, but suddenly it appears in the methodology. They should include what this capacity means in the valorization.

5.      Section 3.1: The authors argue the soluble proteins and bioaccessibility. The reviewer did not see the evidences of the relations between these.

6.     Table 1: In the discussion of these values, they said ratios of proteins etc. in the argument. But this table present the concentration of each components. They need absolute amounts of starting materials, soluble and insoluble fractions to argue these values more effectively and more convincingly.

7.     Table 2: Trp showed up in DSs and DPs. But in DS and DP, Trp was not detected. Where are they from?

8.     Table 3: The figures present mean and standard errors. Why they have different digits? If the mean is XX.X, the error should be y.y, not y.yy.

9.     Discussion: each item in the discussion should be carefully checked if they are given with clear definition. Then the relations between items should be considered to make a good flow of story. There are lots of jumps in the story. Often the arguments are unrelated between the preceding and following sentences. Every jump disrupts the understanding of readers. The similar unclearness are found between paragraphs. It needs fairly extensive revision to make their argument clearer in my honest opinion.

Language itself is fine, but the structure of argument needs more attention.

Author Response

Dear Reviewer, 

We appreciate the comments , which helped to improve the content of the manuscript.

  1. Title: From the title, the unclearness of story is obvious. After reading the manuscript, the reviewer found what they like to say was “The effects of simulated gasto-intestinal digestion on bioactive compounds and prebiotic properties of sunflower and palm kernel meal”. At the beginning, the reviewer thought “how can they digest meals with antioxidant activity? Similar story unclearness are found here and there. The authors need to revise the manuscript to make the story clear to EVERYONE.

The title of the manuscript was constructed to bring a declarative information, with the main result. After the reviewer's suggestion, the following change was made: “Simulated gastrointestinal digestion of sunflower and palm kernel meal present bioaccessible, antioxidant and prebiotic compounds”

  1. Use of abbreviations in the Abstract: The abstract contains the abbreviations such as ORAC, ABTS, DDPH, AAPH, ROS, and so on. Many of them are used without definition. Abstracts should be understandable without other part of manuscript as it can be separately representing the article when they are databased.

Considering this comment, the abstract was modified and the acronyms were abolished as follows:

Sunflower (Helianthus annuus L.) and African palm kernel (Elaeis guineensis Jacq.) are among the most cultivated in the world regarding oil extraction. The oil industry generates a large amount of meal as a by-product, which can be a source of nutrients and bioactive compounds. However, the physiological effects of bioactive compounds present in such matrices are only valid if they remain bioavailable and bioactive after simulated gastrointestinal digestion. This study evaluated the chemical composition, antioxidant, and prebiotic potential of de-oiled sunflower (DS) and de-oiled palm kernel (DP) meal, as well as bioaccessibility of some components before and after in vitro digestion. The DS sample had the highest protein content and the best chemical score, in which lysine was the limiting amino acid. Digested samples showed increased antioxidant activity, measured by chemical in vitro different methods. The digested DS sample showed better antioxidant effect compared to DP, by ORAC, ABTS and DDPH. Moreover, both samples managed to preserve the DNA supercoiling in the presence of the oxidizing agent (AAPH). The insoluble fractions after digestion stimulated the growth of prebiotic bacterium of Lactiplantibacillus plantarum BG112, Lactobacillus acidophilus LA-5, Bifidobacterium animalis BLC1 and Bifidobacterium lactis BB-12, similarly to inulin. In conclusion, simulated gastrointestinal enzymatic digestion promoted in both matrices an increase in protein the bioaccessibility and of bioactives capable of reducing oxidative stress and DNA protection protecting DNA from ROS damage, pointing to a metabolic modulation favorable to the organism.

  1. The story of the Abstract: The story of abstract does not have clear flow. Each item is presented without clarifying the relations between. Meanwhile unnecessary information, such as methods of antioxidant assay, and probiotics tested are included. The author should revise the manuscript with the idea what is important and what are supportive information.

The abstract has been modified and the name of the methods and abbreviation removed.

  1. The story of Introduction: From the third paragraph in the introduction, the authors explain the background of what types of valorization will be handled in this manuscript. After reading the manuscript, items presented here are somewhat unclear for the story. I would rather bring the story to focus on GID effects. They give the emphasis on prebiotics activities, but it was not significant. If the story is focused on how GID can affect, then the story becomes clearer. Moreover, the DNA protective capacity is not discussed here, but suddenly it appears in the methodology. They should include what this capacity means in the valorization.

We are apologizing for review´s comments and have modified the manuscript, according to the following paragraph:

“Phenolic compounds and bioactive peptides are examples of bioactive compounds [8,9]. In order to develop a biological function (bioactivity), the compound must present bioaccessibility, in other words, it must become accessible to be absorbed in the gastrointestinal tract. Therefore, previously, the component needs to be released from the food matrix and became soluble [10]. Phenolic compounds are recognized as antioxidants based on the presence of aromatic rings in their structure. Moreover, their intake is correlated to the reduction of chronic degenerative diseases risk development [8]. A previous study has claimed that the in vitro digestion can increase or reduce the total phenolic compounds content depending on the type of food matrix (liquid or solid), since the dietary compounds at solid ones can protect the phenolics. For example, an increase in total phenolic content was observed in cereals, legumes, vegetables, chocolates, and fruits after the in vitro digestion [8]. Food-derived peptides are sequences of 2 to 20 amino acids, usually inactive in parent protein, that are released by proteolysis of the native protein (for example in fermentative and proteolytic manufacturing processes, in vivo or in vitro digestion) and can develop a bioactive activity, such as lowering blood pressure and cholesterol, being anticancer or antioxidant, among others [9]. Sunflower protein hydrolyses has released antioxidant peptides [11]. However, resistance to simulated digestive processes has not been evaluated.”

  1. Section 3.1: The authors argue the soluble proteins and bioaccessibility. The reviewer did not see the evidences of the relations between these.

As describe in the revised introduction, gastrointestinal process can extract bioactive compounds prior entrapped in the food matrix, that turns to be bioavailable and therefore have the potential to develop some bioactivity, such as the antioxidant capacity.

  1. Table 1: In the discussion of these values, they said ratios of proteins etc. in the argument. But this table present the concentration of each components. They need absolute amounts of starting materials, soluble and insoluble fractions to argue these values more effectively and more convincingly.

The component composition unit was replaced by g/100 g of sample, as well as a small explanation of the results in the table.

“It is important to take into account that composition reflects the centesimal sum of the components, which justifies the increase of some components with the decrease of others”.

  1. Table 2: Trp showed up in DSs and DPs. But in DS and DP, Trp was not detected. Where are they from?

Tryptophan method determination was included in the materials and methods section.

“Tryptophan was determined according to Spies (1967)”. Line 128

Spies, J. R. (1967). Determination of tryptophan in proteins. Analytical Chemistry,

 39(12), 1412e1416.

As well as the explanation that in the determination of total amino acids, tryptophan is destroyed by acid hydrolysis of the protein, being determined by enzymatic reaction, through the enzyme Pronase E. Tryptophan was not determined in the initial samples, probably due to the fibrous complexity of the matrices, which prevented the enzyme from accessing tryptophan (to perform its release and subsequent colorimetric reaction for quantification). In the digested and soluble matrices, tryptophan was more accessible to the enzyme and thus could be quantified by colorimetry. Line 274 - 281

  1. Table 3: The figures present mean and standard errors. Why they have different digits? If the mean is XX.X, the error should be y.y, not y.yy.

The standards errors have been modified according the means.

  1. Discussion: each item in the discussion should be carefully checked if they are given with clear definition. Then the relations between items should be considered to make a good flow of story. There are lots of jumps in the story. Often the arguments are unrelated between the preceding and following sentences. Every jump disrupts the understanding of readers. The similar unclearness are found between paragraphs. It needs fairly extensive revision to make their argument clearer in my honest opinion.
